# Quasi-2D Mn_3_Si_2_Te_6_ Nanosheet for Ultrafast Photonics

**DOI:** 10.3390/nano13030602

**Published:** 2023-02-02

**Authors:** Yan Lu, Zheng Zhou, Xuefen Kan, Zixin Yang, Haiqin Deng, Bin Liu, Tongtong Wang, Fangqi Liu, Xueyu Liu, Sicong Zhu, Qiang Yu, Jian Wu

**Affiliations:** 1School of Transportation Engineering, Jiangsu Shipping College, Nantong 226010, China; 2CAS Key Laboratory of Nanophotonic Materials and Devices & Key Laboratory of Nanodevices and Applications, i-Lab, Suzhou Institute of Nano-Tech and Nano-Bionics (SINANO), Chinese Academy of Sciences, Suzhou 215123, China; 3College of Advanced Interdisciplinary Studies, National University of Defense Technology, Changsha 410073, China; 4College of Science and Key Laboratory for Ferrous Metallurgy, Resources Utilization of Ministry of Education, Wuhan University of Science and Technology, Wuhan 430081, China

**Keywords:** Mn_3_Si_2_Te_6_, magnetism, ultrafast photonics

## Abstract

The magnetic nanomaterial Mn_3_Si_2_Te_6_ is a promising option for spin-dependent electronic and magneto-optoelectronic devices. However, its application in nonlinear optics remains fanciful. Here, we demonstrate a pulsed Er-doped fiber laser (EDFL) based on a novel quasi-2D Mn_3_Si_2_Te_6_ saturable absorber (SA) with low pump power at 1.5 μm. The high-quality Mn_3_Si_2_Te_6_ crystals were synthesized by the self-flux method, and the ultrathin Mn_3_Si_2_Te_6_ nanoflakes were prepared by a simple mechanical exfoliation procedure. To the best of our knowledge, this is the first time laser pulses have been generated using quasi-2D Mn_3_Si_2_Te_6_. A stable pulsed laser at 1562 nm with a low threshold pump power of 60 mW was produced by integrating the Mn_3_Si_2_Te_6_ SA into an EDFL cavity. The maximum power of the output pulse is 783 μW. The repetition rate can vary from 24.16 to 44.44 kHz, with corresponding pulse durations of 5.64 to 3.41 µs. Our results indicate that the quasi-2D Mn_3_Si_2_Te_6_ is a promising material for application in ultrafast photonics.

## 1. Introduction

The pulsed lasers in the communication band (1.5 μm) have an “eye safety” feature and low transmission loss in optical fiber. Owing to the large energy pulses, low cost, and compact structure, the 1.5 μm all-fiber short pulsed lasers have wide and momentous applications in LiDAR [1], remote sensing [2], supercontinuum spectrum [3], space communication [4], and biomedical diagnostics [5]. The passive Q-switching technique is one of the most effective methods to generate pulsed lasers [6,7,8,9]. By applying the Q-switched mechanism, short pulses can be achieved with durations of microseconds (μs) or nanoseconds (ns) and a low repetition rate of kilohertz (kHz) [10,11]. In this technique, the saturable absorbers (SAs) are essential components of fiber laser systems for modulating intracavity loss and generating short pulses [12]. Traditionally, doped crystals, e.g., Co:MALO [13], Cr:YAG [14], etc., are applied to fabricate commercially available passive Q-switching systems. However, the expensive manufacturing process and doping techniques restrict the further development of these crystal-based Q-switchers [15]. In addition, noble metal nanoparticles have also been advocated as SAs for attempting to build pulsed fiber lasers assisted by their localized surface plasmon resonance (LSPR) effect [16,17]. But the aggregation and non-uniformity of nanoparticles could reduce the pulsed laser’s performance.

Over the past decade, numerous studies have been conducted developing two-dimensional (2D) van der Waals (vdWs) materials SAs to meet pulsed laser generation requirements [18], such as graphene [19,20], transition metal dichalcogenides (TMDCs) [21,22], topological insulators (TIs) [23], black phosphorus (BP) [24,25,26], and MXenes [27,28]. These have been proven to have excellent optical properties and produce better laser pulse parameters. Due to the zero band gap, graphene has extremely weak photon absorption [29]. In 2010, Luo et al. used graphene in Er-doped fiber lasers for the first time to realize dual-wavelength pulse output with two wavelengths of 1566.17 nm and 1566.35 nm, respectively [30]. Compared with TMDCs, TIs have a wider wavelength response range, a higher damage threshold, and a greater modulation depth. In 2014, Lee et al. used Bi_2_Te_3_-SA integrated into a holmium-thulium co-doped fiber (THDF) laser to obtain stable pulses [31]. Soon after, BP was proven to have saturable absorption characteristics for the first time. Chen et al. integrated BP into EDF lasers as an SA to obtain pulse output [25]. Analogously, Shi et al. obtained a black arsenic-phosphorus (b-AsP) crystal with an adjustable band gap and designed an atomic ratio by the mineral-assisted chemical gas phase transport method in 2019, transferred it to the fiber facet to prepare a sandwich structure SA, and integrated it into the EDF laser. The output has a pulse width of 7.01 μs [32]. However, the above materials have flaws that limit their future application, such as low modulation depth, environmental instability, or complex preparation methods. Hence, it is urgent to explore novel SAs for pulsed lasers in ultrafast photonics and related applications.

In addition to 2D materials, many new-type low-dimensional nanomaterials as SA potential candidates have also attracted tremendous attention from the research community. For instance, the mode-locking pulse laser based on perovskite CsCu_2_I_3_ microrods SA exhibits ultra-stable properties [33], and the PbS-SA-based EDF pulse laser can realize the switch between Q-switched state and mode-locking state [34]. Moreover, many artificial SAs have also been designed to generate mode-locked pulse lasers, which have outstanding pulse output performance. In the future, artificial SAs may become a real alternative to natural SAs [35]. Because of the nonlinear optical properties of magnetic nanomaterials and their long relaxation times, they have also been proposed to be used as SAs to generate pulsed fiber lasers [36,37]. In 2016, Bai et al. first put forward applying the ferroferric oxide (Fe_3_O_4_) nanoparticles-based SA in pulsed EDF lasers, achieving a laser output wavelength of 1.55 μm and a laser output of 0.8 mW at a pump power of 110 mW, with a maximal pulse energy of 23.8 nJ, a repetition frequency of 33.3 kHz, and a pulse duration of 3.2 μs [38]. In 2018, the Fe_2_O_3_ nanoparticles were synthesized by a co-precipitation method, and then the thin Fe_2_O_3_ polyvinyl alcohol films were prepared to realize pulsed operations in three kinds of fiber lasers due to the broadband polarization-insensitive saturable absorption [39]. Quasi-2D Mn_3_Si_2_Te_6_ nanosheets also exhibit stronger magnetism. Although their ferrimagnetic properties have been extensively studied, their equally interesting optical properties have been very limited. Mn_3_Si_2_Te_6_ is unique in its atomic structure, and the resistivity *ρ*_a_ of the *ab* plane decreases by up to seven orders of magnitude only when the magnetic field *H* is applied along the magnetic hard *c* axis or the saturated magnetic state is not present [40,41]. Besides, the nonlinear optical properties allow for its utilization as an SA in a fiber-pulsed laser.

In this study, the premium Mn_3_Si_2_Te_6_ crystals were prepared by the self-flux method, and the ultrathin Mn_3_Si_2_Te_6_ nanoflakes were adopted as SAs for pulse generation. It has been demonstrated that a stable pulsed fiber laser at 1562 nm can be produced at a low threshold pump power of 60 mW. The maximum power of the output pulse is 783 μW. The repetition rate can vary from 24.16 to 44.44 kHz, with corresponding pulse durations of 5.64 to 3.41 µs. Our results indicate that the quasi-2D Mn_3_Si_2_Te_6_ is a promising material for application in ultrafast photonics.

## 2. Fabrication and Characterization of Mn_3_Si_2_Te_6_

### 2.1. Crystal Growth

By using the standard high-temperature self-flux method, single crystals of Mn_3_Si_2_Te_6_ were grown. A mixture of Mn (powder, 99.95%; Aladdin Chemicals, Shanghai, China), Si (lump, 99.999%; Aladdin Chemicals, Shanghai, China), and Te (lump, 99.999%; Aladdin Chemicals, Shanghai, China) in a molar ratio of 1:2:6 was placed in an alumina crucible, and another empty alumina crucible was kept on top of it with quartz wool separation. All the procedures handling the reagents were done in a glove box filled with highly pure argon gas. To achieve a homogeneous solution, the ampoule was heated to 1273 K in a muffle furnace for 10 h and then dwelled for 24 h. In the following 150 h, the furnace was cooled to 973 K slowly and then remained at 973 K for 24 h so that the crystals could be annealed. The crystals were then separated from the fluxes by centrifuging the ampoule, which had been quickly removed from the furnace. After centrifuging, the black single crystals of Mn_3_Si_2_Te_6_ can be picked out from the remnants in the crucible. In most cases, after cleaning with isopropanol to remove impurities that are not completely crystallized on the surface, high-quality, shiny Mn_3_Si_2_Te_6_ was harvested.

### 2.2. Apparatus and Characterization

Mn_3_Si_2_Te_6_ has a trigonal crystal structure (spatial group No. 163), as shown in Figure 1a [42]. Mn_3_Si_2_Te_6_ consists of MnTe_6_ octahedrons that share edges within the *ab* plane (Mn1 site) and, together with Si-Si dimers, form layers of Mn_3_Si_2_Te_6_. Similar to CrSiTe_3_, which is hexagonal and has a vdWs gap between layers, the layered framework is hexagonal. Nonetheless, Mn_3_Si_2_Te_6_ is composed of layers connected by Mn atoms at the Mn2 site, and these layers fill one-third of the octahedral holes within the vdWs gap to form Mn_3_Si_2_Te_6_ [43]. Importantly, Mn1 has twice the multiplicity of Mn2 in comparison. In the unit cell, the lattice constants are *a* = *b* = 7.029 Å, and *c* = 14.255 Å. The Raman spectra (LabRAM HR Evolution, HORIBA, Villeneuve d’Ascq, France) of Mn_3_Si_2_Te_6_ thin films are shown in Figure 1b, and a 532 nm laser is used as the excitation source in this characterization process. The four characteristic Raman peaks are identified at ~81 cm^−1^, ~100 cm^−1^, ~123 cm^−1^, and ~141 cm^−1^, respectively. This is different from the Raman data reported in the reference [44], and it could be attributed to detecting under different pressures. The characteristic peaks of the Raman spectrum of materials will be affected by the environment in which the materials are located. In ref. [44], the Raman spectrum of Mn_3_Si_2_Te_6_ is measured under high pressure, whereas in this work it is measured under normal temperature and pressure, so there are differences. Figure 1c displays the main XRD spectra (AXS D8 Advance, Bruker, Billerica, MA, USA) characteristic peaks of high-crystallinity Mn_3_Si_2_Te_6_. It matches the rhombohedral structure in space group P-3c1 (PDF # 01-074-1322). The peaks are (004), (006), (008), (00$\bar{1}\bar{0}$), and (00$\bar{1}\bar{2}$) respectively in Mn_3_Si_2_Te_6_. The Mn_3_Si_2_Te_6_ crystal is verified by the (00L) Bragg peaks, and the lattice parameters are determined as *a* = *b* = 7.03 Å and *c* = 14.26 Å by the powder X-ray diffraction of crushed crystals, which is consistent with previous reports [45].

The ferrimagnetic properties of Mn_3_Si_2_Te_6_ crystals are examined by a superconducting quantum interference device (MPMS3, 7T, Quantum Design, USA) in both in-plane and out-of-plane configurations. Figure 2a shows the magnetization *M* data of a Mn_3_Si_2_Te_6_ single crystal when the applied magnetic field *H* lies in the *ab* plane (*H*//*ab*), and that *M* is significantly larger than that when *H*//*c*. We note that there are no remanent moments in either direction, which, in a way, indicates the high quality of the present single crystals. In accordance with this, the magnetization *M* rapidly saturates at *H*//*ab*, reaching ~1.35 μ_B_/Mn at *T* = 2 K. Meanwhile, the zero-field cooling (ZFC) and field cooling (FC) tests at a 200 Oe magnetic field also show the typical ferromagnetic characteristics, as shown in Figure 2b. These data are broadly in line with the initial report about Mn_3_Si_2_Te_6_ [46]. The observed Curie temperature of Mn_3_Si_2_Te_6_ is *T*_C_ ≈ 78 K. These results indicate that the material has distinct magnetic properties.

To study the nonlinear saturable optical absorption properties, the mechanically stripped Mn_3_Si_2_Te_6_ nanosheet is transferred onto the end face of the fiber [47]. The specific dry transfer process is simple and convenient. First of all, the Mn_3_Si_2_Te_6_ bulk was thinned by mechanical exfoliation with blue tape. When a suitable nanosheet has been identified, the underlying fiber end face is fixed on the moving stage. Finally, the tape is pressed against the fiber end face and peeled off very slowly. Figure 3a depicts the end face of the fiber following material transfer. Due to repeated mechanical stripping, in addition to the large nanosheet covering the fiber core, most of the surrounding nanosheets are small in size. The core position is completely covered by Mn_3_Si_2_Te_6_ nanosheets and has been marked by red circles. The optical source of the two-arm detection system [48] is a pulsed laser with an operating wavelength of 1550 nm, a repetition frequency of 16.6 MHz, and a pulse duration of 448 fs.

A simple saturation mode can describe the nonlinear optical absorption
$$T\left(I\right)=1-\Delta R\times \exp\left(-\frac{I}{I_{s}}\right)-T_{ns}$$
where *T*(*I*) is the transmittance rate, Δ*R* is the modulation depth, and *I*, *I_s_*, and *T_ns_* are the input intensity, saturation intensity, and non-saturable absorbance, respectively. Figure 3b takes on the variation curve of nonlinear transmittance with laser energy intensity. By theoretical calculation, the modulation depth and saturation intensity of the Mn_3_Si_2_Te_6_ SA are 17.7% and 16 kW/cm^2^, respectively. It can be seen that the Mn_3_Si_2_Te_6_ SA has a relatively small saturated power density, which also means that saturation can be formed at low power. On the contrary, more power will overflow at high power, which will have an unstable effect on the circulator.

## 3. Generation of the Short Fiber Pulsed Laser

Figure 4 shows the set-up diagram of the pulsed EDF laser that is based on Mn_3_Si_2_Te_6_ SA, where the laser cavity has a ring-shaped design. Through 980/1550 wavelength division multiplexing (WDM), a 976 nm CW laser launched by a laser diode with the maximal power of 400 mW is coupled into the laser cavity as a pump. A 6-m-long EDF (LIEKKI: Er110-4-125) acted as the gain medium. The prepared Mn_3_Si_2_Te_6_ SA is integrated into the laser cavity after the EDF. For unidirectional laser cavity operation, a polarization independent isolator (PI-ISO) is employed. The polarization state of circulating light is optimized using two polarization controllers (PCs). In the experimental operation, one PC always stays still and tunes only the other PC. Additionally, 20% of the pulsed laser power from the cavity is extracted by an optical coupler (OC). The cavity is approximately 11 m long, containing the EDF and tail fibers. A digital oscilloscope (Keysight DSOS104A, 1 GHz, USA), a spectrum analyzer (Yokogawa AQ6370D, Japan), and a power meter (Thorlabs DET08CFC/M, 5 GHz, USA) are utilized to determine the pulse trace, output spectrum, and average output power. The optical fiber is purchased from Yangtze Optical Fibre and Cable Joint Stock Limited Company (YOFC), and the optical fiber devices are purchased from CSRayzer Optical Technology. The illustration in Figure 4 shows the pulse trace at a pump power of 150 mW.

Figure 5a shows the pulse sequences versus different pump powers for the system that operates in a stable Q-switched state between 60 and 160 mW. The Mn_3_Si_2_Te_6_ SA-based EDF pulse laser has high stability because the ferromagnetic material Mn_3_Si_2_Te_6_ is highly stable in the air. Throughout the entire experiment period, even when Mn_3_Si_2_Te_6_ SA is exposed to the air, the pulse sequence remains highly stable. Besides, when the laser is turned off for several hours, we turn it back on, and it still produces a stable pulse laser sequence. Repeated operations can produce a stable pulse. Figure 5b depicts pulse-shaped spectra with soliton-like spectra at various pump powers. The spectrum shows different degrees of broadening. It is necessary to note that if the Mn_3_Si_2_Te_6_ SA is removed by us in the experiment, the Q-switched pulse will not be generated. In addition, the central wavelengths of pulse lasers were always kept steady without drift when the two PCs were tuned simultaneously. It indicates that the PC does not play a leading role and that the number of PCs is not the main reason for the Q-switching phenomenon.

Figure 5c shows the characteristics of a stable single pulse at the pump power of 150 mW, and the corresponding output spectrum of the output pulse as shown in Figure 5d. By varying the pump power, as shown in Figure 5e, the repetition frequency and pulse width modulation ranges of 24.16 kHz to 44.44 kHz and 5.64 µs to 3.41 µs are observed, respectively. Besides, it should also be noted that the pulse duration can be further narrowed by decreasing the cavity length [12] and increasing the modulation depth of the Mn_3_Si_2_Te_6_ SA.

The relationship between the average output powers and pump powers is depicted in Figure 5f. When the pump power is 160 mW, the average output power is up to 783 μW. Once this pump power is exceeded, the Q-switched operation state is destroyed and becomes unstable. This reason can be attributed to the oversaturation of the Mn_3_Si_2_Te_6_ SA and the instability of the laser cavity [25]. Further modifications of cavity settings may further improve laser stability, thereby increasing pulse energy and peak power.

To further evaluate the potential of quasi-2D Mn_3_Si_2_Te_6_ for ultrafast pulse generation, we compare the output performance of Q-switched fiber lasers based on several reported typical SAs (Table 1), such as graphene, black phosphorus (BP), black arsenic-phosphorus (b-AsP), Bi_2_Se_3_ topological insulator (TI), and Fe_3_O_4_ nanoparticles. The results show that quasi-2D Mn_3_Si_2_Te_6_ has a narrower pulse width of 3.41 μs except for Fe_3_O_4_ nanoparticles. The narrow pulse width is the advantage of this Mn_3_Si_2_Te_6_ SA-based EDFL. It should be noted that Fe_3_O_4_ nanoparticles are also a type of magnetic fluid material, and they exhibit the narrowest pulse width of 2.7 μs. Thus, it could be deduced that the magnetic properties of SA materials may help to reduce the pulse width. This point should be further researched in physics in the future. However, the output power of Mn_3_Si_2_Te_6_ SA-based EDFL is far less than that of Fe_3_O_4_ nanoparticle SA-based EDFL. Although the output power of 783 μW based on Mn_3_Si_2_Te_6_ SA is higher than the output power of ~145 μW based on Bi_2_Se_3_ TI SA, the lower output power will have an adverse effect on the practical application of this pulsed laser. So, more efforts should be put into improving the output power of Mn_3_Si_2_Te_6_ SA-based EDFL.

## 4. Conclusions

In summary, a quasi-2D Mn_3_Si_2_Te_6_ nanosheet is successfully prepared, and its ferrimagnetic properties are studied. Meanwhile, a Mn_3_Si_2_Te_6_-based SA is manufactured using a low-cost mechanical exfoliation method, and the nonlinear saturable optical absorption properties in the communication band are investigated. The modulation depth and saturation intensity of Mn_3_Si_2_Te_6_ SA are calculated to be 17.7% and 16 kW/cm^2^, respectively. Then, the prepared Mn_3_Si_2_Te_6_ SA is integrated into a ring-shaped EDF laser cavity to successfully implement the pulsed operation for the first time. The laser pulses at 1562 nm were obtained with stability. The repetition frequency and pulse width modulation range are 24.16 kHz to 44.44 kHz and 5.64 µs to 3.41 µs, respectively, by varying the pump power from 60 mW to 160 mW. The maximal output power is 783 μW. This work certifies the potential of newly quasi-2D Mn_3_Si_2_Te_6_ SA fabricated by low-cost mechanical exfoliation methods for ultrafast photonics applications.

## Figures and Tables

**Figure 1 nanomaterials-13-00602-f001:**
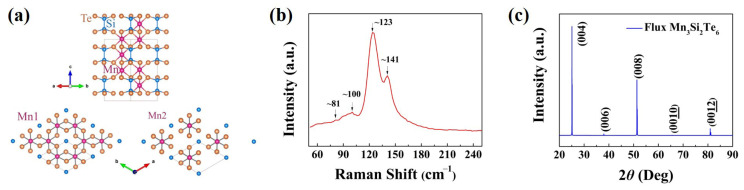
Crystal structure and characterizations: (**a**) crystal structure of Mn_3_Si_2_Te_6_. The Mn atoms are located at the center of the MnTe_6_ octahedra (colored regions), which stack along the *c* axis in honeycomb (Mn1) and triangular (Mn2) layers. (**b**) Raman spectra of Mn_3_Si_2_Te_6_ single crystal at room temperature and 1 atm. (**c**) X-ray diffraction pattern of the Mn_3_Si_2_Te_6_ crystal recorded on the (00L) plane at room temperature.

**Figure 2 nanomaterials-13-00602-f002:**
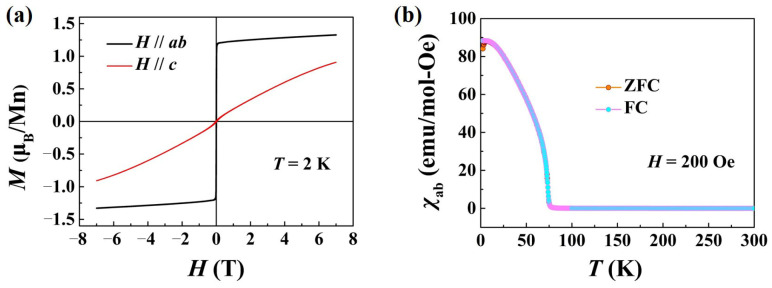
Ferrimagnetic properties of Mn_3_Si_2_Te_6_: (**a**) *M*-*H* curves of Mn_3_Si_2_Te_6_ bulk crystals at *T* = 2 K with magnetic fields along the in-plane (*H∥ab*) and out-of-plane (*H∥c*) directions, respectively. (**b**) ZFC-FC (*H* = 200 Oe, in-plane) curves of Mn_3_Si_2_Te_6_ bulk crystals.

**Figure 3 nanomaterials-13-00602-f003:**
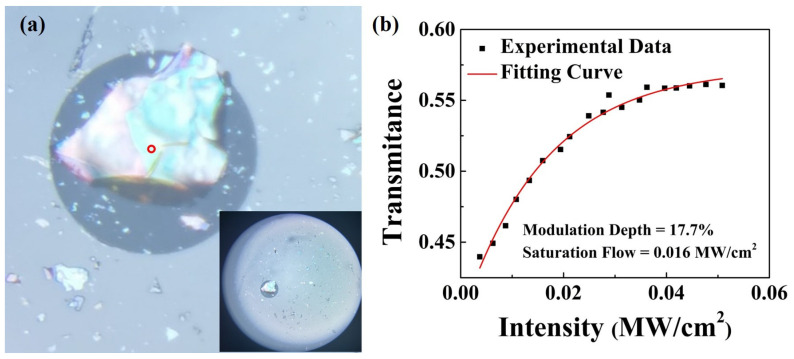
(**a**) Mn_3_Si_2_Te_6_ saturable absorber on the edge of optical fiber. (**b**) nonlinear transmittance of the Mn_3_Si_2_Te_6_ saturable absorber at different light intensities.

**Figure 4 nanomaterials-13-00602-f004:**
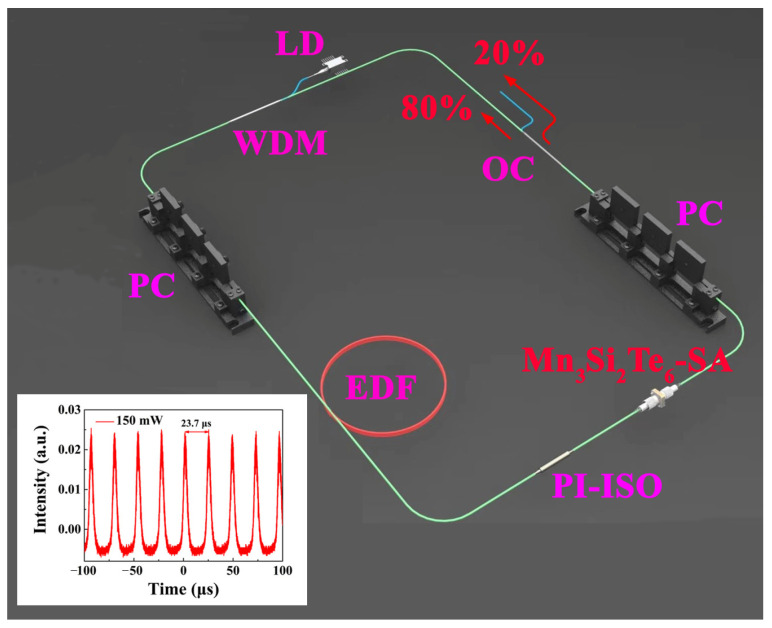
The schematic diagram of the pulsed fiber laser.

**Figure 5 nanomaterials-13-00602-f005:**
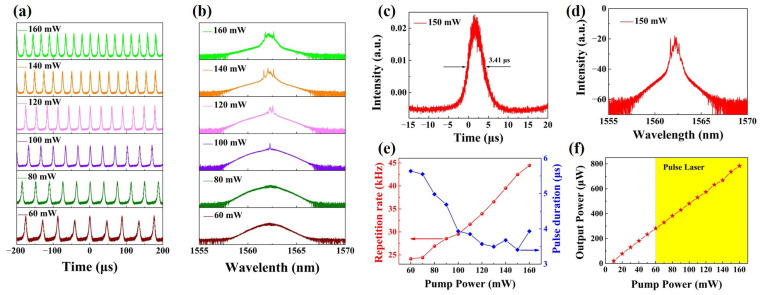
(**a**) pulse sequences. (**b**) the output spectra at given different pump powers. (**c**) the single pulse width at 150 mW pump power. (**d**) the output spectrum of the output pulse at 150 mW pump power. (**e**) repetition frequency and pulse duration as a function of pump power, and (**f**) the relationship between the average output powers and pump powers.

**Table 1 nanomaterials-13-00602-t001:** Technical parameter comparison of several reported passive Q-switched EDFLs based on nanomaterial SA.

Materials	Gain Medium	Wavelength	Pulse Width	Repetition Rate	Pulse Energy	Pump Power	Output Power	Ref.
Graphene	Er-doped	1566.17 nm; 1566.35 nm	3.7 μs	65.9 kHz	16.7 nJ	6.5-82.8 mW	1.1 mW	[30]
BP	Er-doped	1562.87 nm	10.32 μs	15.78 kHz	94.3 nJ	50-195 mW	~1.5 mW	[25]
b-AsP	Er-doped	1559.9 nm; 1560.3 nm	5.26 μs	38.47 kHz	96.4 nJ	20-40 mW	3.68 mW	[32]
Bi_2_Se_3_ TI	Er-doped	1565.14 nm	13.4 μs	12.88 kHz	15 nJ	41.3-84.3 mW	~145 μW	[49]
Fe_3_O_4_ nanoparticles	Er-doped	1562.4 nm	2.7 μs	80 kHz	78.2 nJ	80-342 mW	6.23 mW	[37]
Mn_3_Si_2_Te_6_	Er-doped	1562 nm	3.41 μs	44.44 kHz	-	60-160 mW	783 μW	This work

## Data Availability

Not applicable.

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
