# Peer review of "Quasi-2D Mn3Si2Te6 Nanosheet for Ultrafast Photonics"

_nanomaterials, 2023, doi:10.3390/nano13030602_

Round 1

Reviewer 1 Report

In this paper, Quasi-2D Mn3Si2Te6 nanosheet is fabricated and some characteristics are measured and its ferrimagnetic properties are studied. Then, Mn3Si2Te6-based saturable absorber is fabricated and generation of the short fiber pulsed laser is demonstrated.  I think that the results may be interesting, but I do not think that the provided data is enough to show the usefulness. Also, fabrication process and experimental data has to be explained in more detail.

Author Response

Point 1: In this paper, Quasi-2D Mn3Si2Te6 nanosheet is fabricated and some characteristics are measured and its ferrimagnetic properties are studied. Then, Mn3Si2Te6-based saturable absorber is fabricated and generation of the short fiber pulsed laser is demonstrated. I think that the results may be interesting, but I do not think that the provided data is enough to show the usefulness. Also, fabrication process and experimental data has to be explained in more detail.

Response 1: Thank you for this piece of advice. In order to clearly show the performance of the prepared Mn3Si2Te6 SA-based EDFL, we compare the output performance of Q-switched fiber lasers based on several reported typical SAs, such as graphene, Black phosphorus (BP), Black arsenic-phosphorus (b-AsP), Bi2Se3 topological insulator (TI), and Fe3O4 nanoparticles. The output technical parameters are show in Table 1, which is inserted in the revised manuscript. The results show that quasi-2D Mn3Si2Te6 has the more narrower pulse width of 3.41 μs except Fe3O4 nanoparticles. The narrow pulse width is the advantage of this Mn3Si2Te6 SA based EDFL. It should be noted that Fe3O4 nanoparticle is also a type of magnetic fluid materials, and it exhibits the narrowest pulse width of 2.7 μs. Thus, it could be deduced that the magnetic properties of SA materials may help to reduce the pulse width. This point should be further researched from physics in the future, we cannot temporarily provide more physical results about this deduction in this revised manuscript due to the time constraints. I beg your pardon! The relative discussion about pulse performance have also been increased into the revised manuscript. In the fabrication process, the raw materials are purchased from Aladdin Chemicals. Also, the models of experimental apparatus are added.

Table 1 Technical parameters comparison of several reported passive Q-switched EDFL based on nano-material SA.

Materials

Gain

Medium

Wavelength

Pulse Width

Repetition Rate

Pulse Energy

Pump power

Output power

Ref.

Graphene

Er-doped

1566.17 nm; 1566.35 nm

3.7 μs

65.9 kHz

16.7 nJ

6.5-82.8 mW

1.1 mW

[30]

BP

Er-doped

1562.87 nm

10.32 μs

15.78 kHz

94.3 nJ

50-195 mW

~1.5 mW

[25]

b-AsP

Er-doped

1559.9 nm; 1560.3 nm

5.26 μs

38.47 kHz

96.4 nJ

20-40 mW

3.68 mW

[32]

Bi2Se3 TI

Er-doped

1565.14 nm

13.4 μs

12.88 kHz

15 nJ

41.3-84.3 mW

~145 μW

[49]

Fe3O4

nanoparticles

Er-doped

1562.4 nm

2.7 μs

80 kHz

78.2 nJ

80-342 mW

6.23 mW

[37]

Mn3Si2Te6

Er-doped

1562 nm

3.41 μs

44.44 kHz

-

60-160 mW

783 μW

This work

Reviewer 2 Report

The work submitted for review studies generation properties of a pulsed Er-doped fibre laser using magnetic nanomaterial Mn3Si2Te6 as a saturable absorber. My observations thereupon are presented below:

1. The laser layout contains two fibre-optical polarisation controllers. It is necessary to describe the procedure of their tuning so that the reported results may be reproduced. The tuning procedure should include not only the spool rotation angles, but also the tension used when winding fibre on the spools.

2. Low generation efficiency is demonstrated: at the pumping power of 160 mW, the laser’s output power was as low as 783 µW. Raising the pump power leads to instability of pulsed generation regime. This is an indication that the studied new saturable absorber is not very suitable for pulsed laser generation. It is necessary to provide a critical analysis of the reported results, especially in view of the great number of already studied material-based and artificial saturable absorbers (10.1016/j.yofte.2021.102764, this reference should be added to the manuscript) exhibiting competitive parameters.

3. It is equally necessary to critically evaluate the relatively long duration of the observed pulses (3.41–5.64 μs) and to point out those applications where such long laser pulses may be necessary.

If the Authors address the considerations provided above in their next revision of the manuscript, it may be published in Nanomaterials

Author Response

Point 1: The laser layout contains two fibre-optical polarisation controllers. It is necessary to describe the procedure of their tuning so that the reported results may be reproduced. The tuning procedure should include not only the spool rotation angles, but also the tension used when winding fibre on the spools.

Response 1: Thank you for this question. In the pulse generation experiment, two polarization controllers (PCs) were set into the ring cavity for adjusting the polarization state of circulating light. In the experimental operation, one PC always stays still and tunes only the other PC. If the Mn3Si2Te6 SA is removed by us in the experiment, the Q-switched pulse will not be generated. In addition, the central wavelengths of pulse lasers were always kept steady without drift when the two PCs were tuned simultaneously. It indicates that the PC does not play a leading role and the number of PCs is not the main reason for Q-switching phenomenon. In our previous works, the pulse did not appear when one PC was used, so two PCS were used, so this cavity structure was retained in this experiment. The relative discussion about PCs have been increased into the revised manuscript.

Point 2: Low generation efficiency is demonstrated: at the pumping power of 160 mW, the laser’s output power was as low as 783 µW. Raising the pump power leads to instability of pulsed generation regime. This is an indication that the studied new saturable absorber is not very suitable for pulsed laser generation. It is necessary to provide a critical analysis of the reported results, especially in view of the great number of already studied material-based and artificial saturable absorbers (10.1016/j.yofte.2021.102764, this reference should be added to the manuscript) exhibiting competitive parameters.

Response 2: Thanks for this piece of advice and the recommended literature. The output power of Mn3Si2Te6 SA based EDFL is only 783 µW, which is indeed lower than many typical nanomaterials as shown in Table 1. However, the Mn3Si2Te6 SA based EDF pulse laser have more narrower pulse width and low threshold. It indicates that the Mn3Si2Te6 SA based EDFL is a stable narrow pulse laser. Although the output power of 783 μW based on Mn3Si2Te6 SA is higher than the output power of ~145 μW based on Bi2Se3 TI SA, the lower output power will have an adverse effect on the practical application of this pulsed laser. So, more efforts should be put into improving the output power of Mn3Si2Te6 SA based EDFL. The recommended literature summarizes many artificial saturable absorbers for mode-locking pulse laser generation. In this paper, the Mn3Si2Te6 SA based EDF pulse laser is generated by Q-switching mechanism, so the output performance is inferior to the mode-locking pulse laser. The relative review and discussion have been added in the revised manuscript.

Table 1 Technical parameters comparison of several reported passive Q-switched EDFL based on nanomaterial SA.

Materials

Gain

Medium

Wavelength

Pulse Width

Repetition Rate

Pulse Energy

Pump power

Output power

Ref.

Graphene

Er-doped

1566.17 nm; 1566.35 nm

3.7 μs

65.9 kHz

16.7 nJ

6.5-82.8 mW

1.1 mW

[30]

BP

Er-doped

1562.87 nm

10.32 μs

15.78 kHz

94.3 nJ

50-195 mW

~1.5 mW

[25]

b-AsP

Er-doped

1559.9 nm; 1560.3 nm

5.26 μs

38.47 kHz

96.4 nJ

20-40 mW

3.68 mW

[32]

Bi2Se3 TI

Er-doped

1565.14 nm

13.4 μs

12.88 kHz

15 nJ

41.3-84.3 mW

~145 μW

[49]

Fe3O4

nanoparticles

Er-doped

1562.4 nm

2.7 μs

80 kHz

78.2 nJ

80-342 mW

6.23 mW

[37]

Mn3Si2Te6

Er-doped

1562 nm

3.41 μs

44.44 kHz

-

60-160 mW

783 μW

This work

Point 3: It is equally necessary to critically evaluate the relatively long duration of the observed pulses (3.41–5.64 μs) and to point out those applications where such long laser pulses may be necessary.

Response 3: Thanks for this piece of advice. The Mn3Si2Te6 SA based EDF pulse laser is a stable Q-switched pulse laser rather than a mode-locking pulse laser in this manuscript. Therefore, the pulse width is on the order of microseconds (μs), which is consistent with the theoretical value. As shown in Table 1, all Q-switched EDFL have pulse width on the order of microseconds (μs). The Q-switched pulse laser can be applied in the field of industrial production, such as metal cutting and welding, and optical communication. However, the lower output power of Mn3Si2Te6 SA based EDF pulse laser will have an adverse effect on the practical application of this pulsed laser. So, more efforts should be put into improving the output power of Mn3Si2Te6 SA based EDFL for practical application.

Reviewer 3 Report

The paper refers to application of Quasi-2D Mn3Si2Te6 nanosheet in nonlinear optics. The paper fits the scope of the journal. Experiments were performed properly, however there are some drawbacks of the manuscript:

1.       In 2.2. Apparatus and characterization models of measurements devices and key parameters of characterisation should be added.

2.       Fig. 5 has poor quality. It is difficult to get information. Pictures are small.

3.       Discussion about long-time stability should be added.

Author Response

Point 1: In 2.2. Apparatus and characterization models of measurements devices and key parameters of characterisation should be added.

Response 1: Thanks for this piece of advice. In the revised manuscript, in the part “2.2. Apparatus and characterization”, several descriptions have been supplemented. In this below Table. R1, the suppliers and specifications of all apparatus are described.

Table. R1 The suppliers and specifications of all apparatus

Characterization

Apparatus

Raman spectra

LabRAM HR Evolution, HORIBA, France

XRD

AXS D8 Advance, Bruker, USA

SQUID

MPMS3, 7T, Quantum Design, USA

Digital oscilloscope

Keysight DSOS104A, 1 GHz, USA

Photodetector

Thorlabs DET08CFC/M, 5 GHz, USA

Optical spectrum analyzer

Yokogawa AQ6370D, Japan

Fiber laser devices

Yangtze Optical Fibre and Cable Joint Stock Limited Company (YOFC); CSRayzer Optical Technology

Point 2: Fig. 5 has poor quality. It is difficult to get information. Pictures are small.

Response 2: Thanks for your reminding. We have redrawn Figure. 5 to improve the quality of the image. The resolution of Figure. 5 picture is improved and increasing its clarity.

Point 3: Discussion about long-time stability should be added.

Response 3: Thanks for this piece of advice. The Mn3Si2Te6 SA based EDF pulse laser has high stability because the ferromagnetic material Mn3Si2Te6 is highly stable in the air. Throughout the entire experiment period, even when Mn3Si2Te6 SA is exposed to the air, the pulse sequence remains highly stable. Besides, when the laser is turned off for several hours, we turned the laser back on and still produce a stable pulse laser sequence. Repeated operation can produce a stable pulse. It is a little regrettable that we neglected to record relevant experimental data after verifying its stability through operation. I beg your pardon! The relative discussion has been added into the revised manuscript.

Reviewer 4 Report

The presented study is novel, original, and could bring a significant potential impact on the field of photonics, and it deserves to be considered for publication after revision.

However, besides my agreement with the authors that using magnetic 2D materials for lasing brings significant novelty in the field, the authors fail to link the properties of materials (magnetic and NLO) with observed lasing. At the moment draft looks more like a technical report.

Authors should link the structural properties of the material with observed lasing and significantly extended discussion part (especially regarding physics)

Also, I saw minor typos across the text; authors should carefully check the text before they submit the revision.

Author Response

Point 1: The presented study is novel, original, and could bring a significant potential impact on the field of photonics, and it deserves to be considered for publication after revision. However, besides my agreement with the authors that using magnetic 2D materials for lasing brings significant novelty in the field, the authors fail to link the properties of materials (magnetic and NLO) with observed lasing. At the moment draft looks more like a technical report. Authors should link the structural properties of the material with observed lasing and significantly extended discussion part (especially regarding physics) Also, I saw minor typos across the text; authors should carefully check the text before they submit the revision.

Response 1: Thanks for this piece of advice. This is a very good question, the physics mechanism behind the pulse generation based on magnetic material Mn3Si2Te6 SA is an important research issue. If this issue can be clarified, it will play an important role in the design of magnetic materials based EDFL. In this paper, the magnetic materials Mn3Si2Te6 is fabricated, although their ferrimagnetic properties have been extensively studied, their equally interesting optical properties, such as optical nonlinear property, have been very limited. Therefore, we adopted two-arm detection system to detect the saturable absorption and acquired the lower saturation flow about 16 kW/cm2 relative to other typical nanomaterials. Then, the prepared Mn3Si2Te6 SA was inserted into the ring EDF cavity to generate the Q-switched pulse laser. As show in Table 1, which is inserted in the revised manuscript, we compare the output performance of Q-switched fiber lasers based on several reported typical SAs, such as graphene, Black phosphorus (BP), Black arsenic-phosphorus (b-AsP), Bi2Se3 topological insulator (TI), and Fe3O4 nanoparticles. The results show that quasi-2D Mn3Si2Te6 has the more narrower pulse width of 3.41 μs except Fe3O4 nanoparticles. The narrow pulse width is the advantage of this Mn3Si2Te6 SA based EDFL. It should be noted that Fe3O4 nanoparticle is also a type of magnetic fluid materials, and it exhibits the narrowest pulse width of 2.7 μs. Thus, it could be deduced that the magnetic properties of SA materials may help to reduce the pulse width. This point should be further researched from physics in the future, we cannot temporarily provide more physical results about this deduction in this revised manuscript due to the time constraints. I beg your pardon! The relative discussion about pulse performance have also been increased into the revised manuscript. In addition, the text has been carefully checked for correcting typos.

Table 1 Technical parameters comparison of several reported passive Q-switched EDFL based on nano-material SA.

Materials

Gain

Medium

Wavelength

Pulse Width

Repetition Rate

Pulse Energy

Pump power

Output power

Ref.

Graphene

Er-doped

1566.17 nm; 1566.35 nm

3.7 μs

65.9 kHz

16.7 nJ

6.5-82.8 mW

1.1 mW

[30]

BP

Er-doped

1562.87 nm

10.32 μs

15.78 kHz

94.3 nJ

50-195 mW

~1.5 mW

[25]

b-AsP

Er-doped

1559.9 nm; 1560.3 nm

5.26 μs

38.47 kHz

96.4 nJ

20-40 mW

3.68 mW

[32]

Bi2Se3 TI

Er-doped

1565.14 nm

13.4 μs

12.88 kHz

15 nJ

41.3-84.3 mW

~145 μW

[49]

Fe3O4

nanoparticles

Er-doped

1562.4 nm

2.7 μs

80 kHz

78.2 nJ

80-342 mW

6.23 mW

[37]

Mn3Si2Te6

Er-doped

1562 nm

3.41 μs

44.44 kHz

-

60-160 mW

783 μW

This work

Round 2

Reviewer 4 Report

In the rebuttal letter, the authors answer the referee's comments. I am not completely satisfied with the lack of a deep NLO study, but I understand that the proposed research can not be done in a short time frame. 

In the end, based on the rebuttal letter and the study's novelty, I recommend publishing the revised draft in the unchanged form.